# Essential Topics for the Regulatory Consideration of Phages as Clinically Valuable Therapeutic Agents: A Perspective from Spain

**DOI:** 10.3390/microorganisms10040717

**Published:** 2022-03-26

**Authors:** Roberto Vázquez, Roberto Díez-Martínez, Pilar Domingo-Calap, Pedro García, Diana Gutiérrez, Maite Muniesa, María Ruiz-Ruigómez, Rafael Sanjuán, María Tomás, María Ángeles Tormo-Mas, Pilar García

**Affiliations:** 1Department of Biotechnology, Ghent University, 9000 Ghent, Belgium; rvazqf@gmail.com; 2Telum Therapeutics SL, 31110 Noáin, Spain; rdiez@telumther.com (R.D.-M.); dgutierrez@telumther.com (D.G.); 3Institute for Integrative Systems Biology, University of Valencia-CSIC, 46980 Paterna, Spain; pilar.domingo@uv.es (P.D.-C.); rafael.sanjuan@uv.es (R.S.); 4Center for Biological Research Margarita Salas (CIB-CSIC) and Centro de Investigación Biomédica en Red de Enfermedades Respiratorias (CIBERES), 28040 Madrid, Spain; pgarcia@cib.csic.es; 5Department of Genetics, Microbiology and Statistics, University of Barcelona, 08028 Barcelona, Spain; mmuniesa@ub.edu; 6Internal Medicine, Infectious Diseases Unit, Hospital Universitario 12 de Octubre, 28041 Madrid, Spain; rryruiz@gmail.com; 7Department of Microbiology, Hospital Universitario de A Coruña (INIBIC-CHUAC, SERGAS), 15006 A Coruña, Spain; ma.del.mar.tomas.carmona@sergas.es; 8Study Group on Mechanisms of Action and Resistance to Antimicrobials (GEMARA) on behalf of the Spanish Society of Infectious Diseases and Clinical Microbiology (SEIMC), 28003 Madrid, Spain; 9Spanish Network for Research in Infectious Diseases (REIPI), 41071 Sevilla, Spain; 10Centro de Investigación Biomédica en Red en Enfermedades Infecciosas (CIBERINFEC), Instituto de Salud Carlos III, 28029 Madrid, Spain; 11Severe Infection Group, Hospital Universitari i Politècnic La Fe, Health Research Institute Hospital La Fe, IISLaFe, 46026 Valencia, Spain; tormo_man@iislafe.es; 12Dairy Research Institute of Asturias, IPLA-CSIC, 33300 Villaviciosa, Spain; 13DairySafe Group, Health Research Institute of Asturias (ISPA), 33011 Oviedo, Spain

**Keywords:** phage therapy, bacteriophages, endolysins, antimicrobial resistance, compassionate use, drug regulation

## Abstract

Antibiotic resistance is one of the major challenges that humankind shall face in the short term. (Bacterio)phage therapy is a valuable therapeutic alternative to antibiotics and, although the concept is almost as old as the discovery of phages, its wide application was hindered in the West by the discovery and development of antibiotics in the mid-twentieth century. However, research on phage therapy is currently experiencing a renaissance due to the antimicrobial resistance problem. Some countries are already adopting new ad hoc regulations to favor the short-term implantation of phage therapy in clinical practice. In this regard, the Phage Therapy Work Group from FAGOMA (Spanish Network of Bacteriophages and Transducing Elements) recently contacted the Spanish Drugs and Medical Devices Agency (AEMPS) to promote the regulation of phage therapy in Spain. As a result, FAGOMA was asked to provide a general view on key issues regarding phage therapy legislation. This review comes as the culmination of the FAGOMA initiative and aims at appropriately informing the regulatory debate on phage therapy.

## 1. General Aspects of the Use of Phages as Antimicrobials

The emergence of multi-drug-resistant (MDR) bacteria seriously undermines our ability to control bacterial infectious diseases. A recent study shows that MDR pathogens already cause more than one million deaths a year, and the prospects are even more concerning [1,2]. Therefore, short-term proactive measures are urgently needed. Proposed strategies include controlling the spread of antibiotic resistance, designing new antibiotics, and developing alternative therapies. Bacteriophages (phages), bacterial viruses, were discovered in the second decade of the 20th century and soon used to treat bacterial infectious diseases with encouraging results [3]. However, phage therapy was abandoned due to the discovery of antibiotics, with the exception of the Soviet Union and some Eastern European countries (due to hindered access to antibiotics there). Recently, and in the face of increasing bacterial resistance to antibiotics, phages have re-emerged as alternative or complementary therapies to control bacterial infections [4]. The advantages and disadvantages of phage therapy are discussed below.

### 1.1. Advantages

(a)Activity against antibiotic-resistant bacteria. Phages can infect and kill bacteria, including MDR strains [5]. This is the most obvious advantage towards recovering phage therapy to fight antimicrobial resistance today. Moreover, the composition of the phages for therapy may be designed to impose an evolutionary trade-off in which the evolution of bacterial resistance to phage results in increased susceptibility to antibiotics [6]. Under this rationale, the combination therapy of phages plus antibiotics has a remarkable potential to smartly tackle antibiotic resistance by both eliminating resistant strains and preventing the dissemination of resistance genes [7,8]. Many works have been published thus far where phage–antibiotic synergy, known as PAS effect, is reported. This suggests that combined therapy can be the safest and a more advantageous approach, as it minimizes resistance and virulence [9,10,11].(b)Specificity. The high specificity of many phages towards their host bacterial strains makes them a highly selective therapy that prevents the dysbiosis of the healthy microbiota. Contrary to antibiotics, only strains of the same genus or species—and often just one or very few strains within a species—are susceptible to infection by a given phage, protecting the normal microbiota and reducing side effects [12]. The specificity of phages lies in the bacterial receptor that the virions recognize through one or more receptor-binding proteins [13], and can be further driven by post-entry anti-phage defense mechanisms [14]. Phages can be polyvalent (i.e., with a broad host range) if they recognize a receptor present in several bacteria or, alternatively, their specificity can be very restricted if they bind receptors exclusive to a single bacterial type. Another possibility is that the phage uses a receptor that is only expressed under certain conditions which therefore restricts its infectivity (e.g., the receptor of phage lambda is the maltose receptor, which is only expressed in the presence of maltose [15]).(c)Multiplication at the site of the infection (auto-dosage). Phages can multiply at the site of the infection. Once the phages reach the targeted bacteria, they will replicate and generate progeny. Therefore, if sufficient phage particles are able to reach the infection site, phage therapy can be considered as auto-dosage treatment. Furthermore, once the infection is successfully controlled, phages would be eliminated in the absence of bacterial hosts. Thus, whenever an auto-dosed, “active” treatment is achieved, it can elicit the infection eradication by only a single administration [16].(d)Ubiquity and diversity. Phages can be found in virtually any environment [17], and they play an utmost important role in ecosystems by regulating bacterial populations [18], including the human microbiota. The main practical consequence of such ubiquity and the concomitant diversity is the ease of discovery of novel phages, which contrasts with the currently slow antibiotic discovery rate.(e)Evolvability. Phages are evolving entities and, therefore, can be optimized using directed evolution techniques. This opens up many possibilities compared to conventional treatments, which are stable chemical compounds. Phage evolvability can be exploited in many ways, such as increasing lytic capability, improving particle stability, expanding the host range, or counteracting bacterial resistance. For instance, the Appelmans’ protocol uses spontaneous mutation and recombination among phages present in a cocktail to produce phage variants capable of infecting initially non-susceptible bacterial strains [19,20].(f)Safety. Humans are carriers of many different phages forming the phageome [21,22]. Their biological functions, beyond regulating bacterial populations, are not yet entirely clear [23]. However, their widespread presence in the human body seems to be a good safety indicator. Moreover, there is evidence of phage safety from clinical trials and intake of phage-treated foods [24]. A potential concern of phage therapy could be the release of bacterial endotoxins after lysis of the targeted bacterial cells. It should be noted, however, that similar observations have been made regarding conventional therapy with certain antibiotics [16], and that the current literature does not support detrimental inflammatory reactions upon phage administration. Phages may also enter tissues that are not the specific target of the treatment, but these interactions also do not appear to produce side effects [25].

### 1.2. Weaknesses

(a)Phage-resistance. In the same way that resistance to antibiotics emerges, bacteria can become resistant to phage infection. The most common solution to address this involves the use of cocktails of different phages, rather than a single phage, and/or the “à la carte” selection of phages for each particular infectious isolate. This makes it much less likely that the host will become resistant to all phages at the same time [26]. The so-called “step-by-step” technique is an interesting approach in which phages are isolated against phage-resistant bacterial mutants in successive screening steps to obtain other phages capable of infecting resistant variants. By this method, the natural antagonistic co-evolution that would occur upon treatment is mimicked prior to therapy, thus generating a phage cocktail able to infect both the original bacterium and the foreseeable resistant variants [27]. Moreover, the emergence of phage-resistant bacteria is not always a disadvantage, since it sometimes involves a decrease of the fitness or virulence of the bacterial host [28], or may resensitize bacteria to antibiotics [9].(b)Specificity. This feature can be a double-edged sword. Phage specificity requires careful susceptibility testing of each bacterial pathogen before treatment, which may be viewed as an issue for certain acute infections that require urgent action. In addition, this specificity may require either the development of large phage libraries and/or extensive sampling and screening efforts to provide sufficient coverage of bacterial diversity. This can be a daunting task and has posed major regulatory issues, since, according to the current framework, each individual phage should undergo review and approval. In addition, the eventual need to develop a different phage preparation for each bacterial pathogen, as a personalized medicine, reduces business profitability and can be viewed as a serious drawback by pharmaceutical companies. Again, phage cocktails targeting different receptors or different bacterial strains would be a potential solution.(c)In vivo phage activity. There is not necessarily a correlation between the in vitro and in vivo behavior of a phage, particularly regarding its propagation ability. This is due both to the complexity of body fluids and the ecological in vivo interactions [29,30]. In addition, the phage propagation is dependent on the physiological state of the bacterial host, which may not be optimal for infection in vivo (for example, depending on whether the bacterium is embedded or not within a biofilm, the expression of receptor molecules, etc.) [31]. Moreover, phages are bigger than antibiotics and, therefore, diffuse less efficiently. This limitation is aggravated in vivo, where multiple physical barriers are encountered. Therefore, the probability of infection at low phage and bacterial densities is low, and the threshold densities required to ensure phage infection may often require the administration of very high phage doses [32,33].(d)Immune response. Since phages are made of biomacromolecules, they are potentially capable of eliciting an immune reaction upon administration [34]. Generally, the immune reactions against phage components are not considered problematic for the individual under treatment, although they do have a relevant contribution to the outcome of phage therapy [35]. On one hand, the immune response potentially causes the removal of phages from the system [36], although this effect may be overcome by adjusting such parameters as dosing, administration route, etc. On the other hand, synergism with the immune antibacterial response seems relevant for therapeutic success [37], although some evidence suggests that phage therapy can also be successfully applied in immunocompromised patients [38]. To sum up, the interaction of phages with the immune system is complex and not yet well understood, although many unknown implications seem to affect the therapeutic efficacy without contradicting the presumed safety of phage therapy.(e)Gene transfer. Phages potentially have the ability to modify the genome of the host bacteria, which may increase their virulence or dissemination of antibiotic resistance genes [39]. Indeed, phages can mobilize large fragments of bacterial genomes at relatively high frequencies [40,41]. To date, it is not known whether the mobilized DNA is randomly selected or whether the transfer of some particular genes is favored, e.g., those related to virulence, survival, or fitness of the host strain. A relevant mechanism for phage-mediated transfer of particular genes is associated with lysogeny (the integration of the phage genetic material into the bacterial chromosome) [42]. Therefore, this issue could be minimized by selecting exclusively virulent phages, as well as by analyzing phage genomes in detail to ensure that they do not contain genes encoding toxins or any other undesirable genes.

## 2. Obtaining Therapeutic Phage Preparations

The main steps for obtaining phage suspensions suitable for use in clinical settings are summarized in Figure 1. 

### 2.1. Selection of Screening Host Strains and Phages Intended for Therapy

The process of phage therapy begins with the identification and selection of phages and bacterial strains suitable for phage production. Phages are found in any environment, even under extreme conditions, being the most abundant biological entities on our planet, with a current total estimate of 10^31^ particles [17]. Phage “hunting” site selection will depend on the bacteria to be eliminated [43], since phages infecting a certain host can normally be isolated from environments inhabited by such bacterium. However, it may be difficult to find phages active against a bacterial isolate from samples of the same patient, since the resident bacteria (including the pathogenic one) are probably lysogenic, tolerant, or resistant to the accompanying phages. Therefore, alternative sampling places are recommended. Many pathogenic bacteria also present free-living variants in nature, making it possible to find suitable phages for therapy in environmental samples. Human-made sites or infrastructures, such as wastewater treatment plants, usually contain a high diversity of phages active against human pathogens [44].

Once samples are collected, phage screening parameters will depend on the target bacterium. In general, phages can be isolated using reference strains or clinical isolates as hosts for propagation [45]. In some cases, the bacterium of interest may not be amenable to culture, and thus a surrogate host may be used instead (e.g., *Mycobacterium smegmatis* is strongly preferred for the isolation of mycobacteriophages over the pathogenic, slowly growing *Mycobacterium tuberculosis* [46]). If a broad-range phage is desired (for example, one that targets several strains), isolation with multiple bacterial strains may be preferred, although single strain enrichment does not necessarily preclude finding broad-range phages [43]. Apart from the search for new phages, it is also possible to use those that are already available in laboratories around the world, or in phage banks [47,48]. In general terms, the isolation of phages does not pose a great difficulty neither methodologically nor economically, as long as its host bacterium is culturable. Indeed, phage isolation and selection methods are as old as the first descriptions of phages and are still considered as solid and useful protocols nowadays.

### 2.2. Small- and Large-Scale Production Processes

Phage production is typically carried out by the double-layer agar method, which allows isolation of individual phages, and then liquid culture of single plaques [49]. A pure phage lysate (containing a single type of lysis plaques) is obtained by centrifugation, filtration, and/or further purification methods (see below). The process can be scaled up and optimized to an industrial level by using bioreactors of different sizes [50], which allow continuous and semi-continuous production. The latter seems the most advantageous for large-scale standardized production, since it avoids co-evolution of the phage with the bacteria (although it may be operationally complex) [51].

### 2.3. Purification of Phage Solutions

The main goal of this process is to avoid the presence of bacterial toxins, lipopolysaccharide, or other cellular debris, in the phage suspensions [52]. In addition to centrifugation and membrane filtration, it may be necessary to apply additional steps, such as dialysis, ultrafiltration, or treatment with organic solvents. To verify the absence of toxic components, the corresponding tests should be performed and, if necessary, additional purification methods such as specialized filtration, affinity chromatography, tangential flow filtration, and/or CsCl gradient ultracentrifugation can be used. An alternative may be anion exchange chromatography, as demonstrated using Convective Interaction Media^®^ (CIM) columns [53]. These more specialized approaches often require laborious optimization for each phage (or groups of related phages), but also allow the concentration of viral particles in lysates with low phage titers.

### 2.4. Storage

Depending on the phage, phages can be stored at different temperatures, commonly at 4 °C, −80 °C, or in liquid nitrogen (−196 °C), or can be freeze-dried [54,55]. Protection of lysates against evaporation or contamination is often sufficient to minimize the decrease in their titer over time. Additives can be added to prevent or delay the loss of infectivity. Mg^2+^ and Ca^2+^ ions (around 10 mM in the form of CaCl_2_ or MgSO_4_) are the most used supplements since they are added to the culture medium before infection to facilitate adsorption and are then present in the recovered lysate [54,56]. Other common additives are cryopreservants such as disaccharides (lactose, sucrose, trehalose) or polyethyleneglycol, as well as gelatin or Ficoll [57,58]. Adsorption of phages to cell debris can cause a considerable decrease in titer, implying that removing cellular contaminants from a crude lysate is also important for storage. In addition, minimizing the number of passages is an important issue since multiple rounds of propagation combined with the high mutation frequencies shown by phages can lead to genetic differences between the original isolate and its progeny [54].

### 2.5. Formulation and Administration

Phages are basically protein structures and, therefore, they are susceptible to proteases, certain chemical compounds, high temperatures, pH, and ionic strength. Thus, it is important to use an appropriate formulation to ensure that the phage titer remains stable both during formulation and storage and in the in vivo environment where they are applied. Again, the optimal formulation conditions may vary depending on the specific phage, so this deserves careful consideration in the case of phage cocktails, as each type of phage may require individually tailored conditions [59]. However, the major determinant of phage therapy formulations is the method chosen for delivery, which, in turn, depends on the infection site:

(a)Oral administration is appropriate for gastrointestinal infectious diseases. In some cases, oral phages given without additional protection, as water-based liquid suspensions, reportedly survived gastric passage and were recovered in the feces [60,61,62]. The formulation efforts for liquid phage suspensions are typically minimal since phages are just prepared in sterile buffers such as phosphate-buffered saline (PBS), the bacterial growth medium, standard saline, or water [61,63,64,65]. More elaborate formulations specifically intended for oral administration can improve phage survival through the extreme conditions of the gastrointestinal tract. For example, encapsulation protects phages from the highly acidic stomach environment and digestive enzymes [66]. Furthermore, their release can be triggered in a controlled manner, for example, pH-dependent release, with capsules programmed to become permeable at different pHs regarding the aimed site of action: from the stomach (pH 1–3) to the small intestine (pH 5.5–6.5) or the colon (pH 6.5–7.2) [67]. A wide range of natural and synthetic polymers are available that offer considerable plasticity for tailoring phage encapsulation and subsequent release to different biomedical applications, including polysaccharides, natural or synthetic plastic polymers, liposomes, and micelles [68,69].(b)Topical administration of phages is chosen for skin infections, wounds, burns, ulcers, and osteoarticular infections [70]. Phages have been topically administered in liquid, semi-solid, and liposome-encapsulated formulations, as well as phage-immobilized wound dressings [63]. When using liquid preparations, they may just be dripped onto the infected site or applied in a gauze soaked with the preparation. Alternatively, gel or cream formulations are suitable to overcome some of the limitations of liquid preparations, with a preference for hydrogels over organic solvent-based gels. This is especially relevant for the treatment of burn wounds, since hydrogels help keep the wound hydrated as much as they favor phage stability [63]. Commercial infection-care products can also be used as a formulation basis for topically delivering phages, but care should be taken as to whether the composition of the product reduces phage infectivity [71].(c)The local phage treatment of respiratory infections requires preparing phages either as stable liquid formulations for intranasal instillation or nebulization, or as a solid powder in an inhalable form [72]. The most popular formulations for respiratory infections are liquid suspensions, due to the relative simplicity of preparation. Nebulization of liquid phage suspensions has been tested with mixed outcomes, generally suggesting that temperature, relative humidity, the nebulization-induced mechanical stress, delivery efficiency of the system, and the nature of the phage itself greatly influence the outcome. Regarding dry powder inhalation, the methods to obtain solid phage formulations include freeze-drying or spray-drying. In general, both processes subject phages to diverse stresses that may impact their infectivity [73], but the control of key parameters and addition of suitable excipients, including polymers for encapsulation, can enhance phage preservation [74,75].(d)Intravenous administration is recommended in the treatment of systemic infections [76]. In this case, liquid phage suspensions are typically used, normally prepared in aqueous buffers suitable for inoculation [64,72].(e)Intravesical instillation of liquid phage preparations has also been used to treat genitourinary tract infections [77,78].

## 3. Quality Criteria for Therapeutic Phage Preparations

As products intended for human therapy, phage preparations must comply with certain quality criteria that ensure their safety for clinical use, as well as exhaustive traceability documentation. The production and delivery parameters must be set up by the phage preparation supplier in accordance with the applicable regulations (see Section 4) and to meet the agreed quality attributes. For phage-based products, these quality controls would typically include:(a)Phage identity. The identity of each phage is defined by its specific genomic sequence [79,80]. Metagenomics has already been proposed as a quality control method for some vaccines [81], and thus has also been used to assess the composition of commercial phage products [82,83]. This method allows the detection of biological contaminants while also assuring the active product identity. While random mutations during propagation are inevitable, they need to be as limited as possible by process design (e.g., minimizing subcultivation steps), and functional properties should be regularly tested with validated quality controls, as even single-nucleotide polymorphisms can lead to significant phenotypic changes. However, a highly discriminating PCR-based genotyping technique might be sufficient in some cases [79]. The maximum acceptable level of genomic divergence between the master batch and the phage population in the therapeutic product, as well as the frequency of such quality check, should be nonetheless adjusted on a case-by-case basis [79].(b)Phage Titer. The titer of each individual phage is classically assessed by the double-layer agar method. An alternative is lethality curves, in which the kinetics of phage-induced lysis are assessed by measuring the optical density of phage-infected bacterial cultures [84]. Other methods, such as qPCR and ELISA, can be used to quantify phages, but they do not necessarily quantify infectious viral particles, whereas double-layer and lethality assays do determine biological activity [79].(c)General Purity. For biopharmaceuticals, the purity and correct composition is classically assessed by high-performance liquid chromatography, combined with mass spectrometry if necessary. These methods can be used to identify phage capsid proteins, toxins, or other bacterial proteins. Because of the potential risk posed by the necessary production with pathogenic bacterial hosts, quality criteria should specify maximum levels for contaminants such as toxins or bacterial DNA, which normally must be tested with specific and appropriate molecular biology methods as specified below.(d)Toxins. Several in vitro methods have been developed for endotoxins quantification: gel-clot, turbidimetric, and chromogenic tests. Among the latter, the limulus amebocyte lysate assay is the most widely used [85]. When this assay is not applicable, e.g., due to masking effect, a reporter cell line can be used [86]. In addition, several commercial assays can be used to detect other toxic bacterial proteins, including ELISA or assays based on reporter cell lines.(e)Contaminating Nucleic Acids. Quality controls may also be required to determine the concentration of contaminating nucleic acids (i.e., non-phage nucleic acids). The presence and concentration of residual nucleic acids are typically checked by qPCR.(f)Other Quality Controls. Current regulations on sterility or general quality parameters in pharmaceutical products should also apply to phage-based pharmaceuticals [87]. Some parameters that may need to be checked are the total microbial load, pH, osmolarity, visual appearance, and/or maximum water content (in lyophilized preparations) [79].

## 4. Regulation for Phage Preparations

Perhaps the greatest hurdle to the implementation of phage therapy in Western medicine is the lack of appropriate regulation. If phage preparations are considered “classical” medicinal products, they must comply with the corresponding legislation concerning medicinal products production and quality. This essentially means that they should follow GMP (good manufacturing practices) requirements, which has imposed an important problem for the development of medicinal phage preparations in terms of increasing costs or complicating management for developing large-scale production [88].

Although therapeutic phages shall be considered indeed a medicinal product, the nature itself of the phage makes it essentially different from common antimicrobial chemotherapy. Indeed, phage specificity, virus–host co-evolution, or a complex in vivo pharmacological behavior have negatively influenced the outcome of many of the phage therapy clinical trials conducted in contemporary times [33,89,90,91]. Therefore, the clinical trial path to the market still seems rather far away. A provisional conclusion has been that the parameters by which we evaluate phage therapy should perhaps be specifically adapted and may not be the same as those of chemotherapeutics [92]. So far, the most successful approach to applying phage therapy in the clinic has been tailored formulations, i.e., phage preparations specifically designed and developed to tackle the infection present in a specific patient. In fact, the compassionate use of phages intended for specific patients without better treatment options is the current regulatory framework for phage therapy in most countries [4,88]. Within this framework, some of the stricter requirements can be bypassed, although it still requires approval from the competent authorities in every individual case.

A recently adopted strategy in Belgium has allowed a more systematic and practical approach to personalized phage therapy, which also provides production and handling flexibility (e.g., by not forcing strict GMP compliance). The key to this approach is to consider phage products as magistral preparations, since, unlike medicinal products, these are subject to less strict regulation in terms of their production and marketing [93,94]. In this context, phages are regarded as active pharmaceutical ingredients (APIs) (substances used in a finished pharmaceutical product intended to procure a pharmacological effect), which are provided with an external quality assessment according to a dedicated monograph. A practical infrastructure to operate under this regulatory framework would be a dedicated phage bank in which each conserved phage should be certified to be used as an API in such a way that the certification covers all essential quality attributes (including a “genetic passport” issued for each phage in particular) [48,94]. In Spain, magistral formulations are governed by European regulations and by the Royal Spanish Pharmacopoeia (order SSI/23/2015, BOE-A-2015-467) and by the consensus document specifying the requirements for APIs for use in magistral formulations (CTI/FM/150/02/16) and following the rules of correct elaboration and quality control (RD 175/2001, BOE-A-2001-5185). These documents define “magistral formulation” as “the drug intended for an individualized patient to explicitly fulfill a detailed medical prescription of the active ingredients that it includes, prepared by a pharmacist or under his direction according to the rules of proper preparation and quality control established for this purpose, dispensed in a pharmacy or pharmaceutical service and with due information to the user”. In the case of phages, express authorization from the competent agency (in Spain, the AEMPS) would be required to use them in magistral formulations, as well as a dedicated monograph provided by the supplier and stating the “rules of proper preparation and quality control”. The exemplary monograph elaborated in Belgium can be accessed as Supplementary Material in reference [94], and it contains, in a summarized manner, most of the issues covered by this review. While this bold framework may be practical for the time being, it has not yet been adopted by many countries. To our knowledge, there are no examples in Spain of phage therapy application under this regulatory framework, although phages have been administered as compassionate medicines. In any case, it is altogether clear that phage therapy may not fit within the traditional drug regulations. Some calls have already been made to adapt or develop a specific regulation for phage therapy [95,96]. The main objectives of such a regulation should be, at least, (a) developing adequate criteria for assessing the quality, efficacy, and safety of phage products; (b) lightening the formal procedures for administrating personalized phage therapy; and (c) fairly distributing responsibility and compensation among the involved actors.

Regardless of the sub-optimal regulatory status of phage therapy in most countries, including Spain, phage preparations for therapy are available either commercially (e.g., through the Eliava Institute in Georgia, which keeps and promotes the Eastern tradition in phage therapy) or through the request to academic or clinical institutions devoted to phage therapy research and promotion.

## 5. Clinical Trials and Prospects for Phage Therapy

Although potentially controversial [97], an adapted regulation in support of personalized phage therapy (perhaps following the lead of the Belgian experience) can be a shortcut to bring phage therapy to the clinic and even to the market in Spain. However, as it has been pointed out, once phage therapy becomes increasingly adopted in clinical practice and/or the number of untreatable infections begins to rise, the personalized phage therapy would need such infrastructure dedication that it may become impractical (unless healthcare/research systems are decisively funded to meet the need). This situation would surely favor a more traditional, market-based approach [89]. The latter would still require phage formulations to be developed through the common drug clinical pipeline, i.e., demonstrating efficacy in randomized clinical trials. In fact, one of the major hurdles to the practical prospects of phage therapy is the lack of substantial evidence within the current clinical trial standards. While a good amount of mostly successful case reports has been recently published, the phage research community still fails to deliver convincing randomized clinical trial results supporting phage therapy efficacy. If we consider the two better-known recent phage therapy trials that yielded disappointing results (i.e., the PhagoBurn trial and the Nestlé Bangladesh diarrhea trial), available evidence suggests a poor understanding of the complex phage behavior in vivo, as well as improper assumptions on the pathogenic bacteria susceptibility [33,90]. Both issues greatly complicate standardized, large-scale trials. For example, the abovementioned trials reported a lower phage dose than expected at the infection site and resistance to the cocktails among the infecting bacteria. A posterior reflection on the Bangladesh study by one of its authors pointed out that the polymicrobial nature of some infections makes them a tough target for phage therapy alone, and even more complex for a productive randomized controlled trial [89]. Therefore, a possible suggestion towards successful clinical trials may be to exclusively target well-characterized infections, dedicating extra care to consider the appropriateness of the targeted pathologies [98]. In actuality, the only phase 2 clinical trial that has produced positive efficacy results to date followed this path, by selecting a single-pathogen infection (chronic otitis caused by antibiotic-resistant *Pseudomonas aeruginosa*) and testing the identity and quantity of bacteria present in every patient [99]. Further issues regarding phage production (according to GMP), storage, product shelf-life, dosage, and proper formulation/administration need to be exhaustively assessed and optimized to ensure the future success of clinical trials. Regardless of the potential hurdles, many phage clinical trials are currently ongoing at different stages, both including predefined phage cocktails and personalized interventions, thus safeguarding a potential future regularization of phage therapy based on clinical trial evidence. Some examples of ongoing clinical trials are included in Table 1.

## 6. Most Urgent Indications for the Application of Phage Therapy in Spain

In our opinion, the use of phage therapy would be advisable:(a)In the case of a severe infection produced by an MDR bacterium.(b)When the infection occurs in an area reluctant to the use of antibiotics, such as in prosthetics.(c)Or, in general, whenever there is no standard of care option available, such as patients suffering from hypersensitivity to the antibiotic treatment.

In these situations, i.e., in the absence of any other satisfactory therapy and when there is a real risk to the patient’s life or significant deterioration of their quality of life, the use of phages, possibly in combination with standard-of-care antibiotics, would be advisable. Conversely, whenever the patient’s infection can be satisfactorily treated with antibiotics, those will always be the therapy of choice. Additionally, a proper diagnostic of the infection-causing bacteria must be conducted (see Section 5), as well as a susceptibility proof of such bacteria towards the phage preparation to be applied (what is known as a “phagogram”) [100,101]. We provide details below on the most urgent indications in which a regularization of phage therapy could have a greater impact.

### 6.1. Cystic Fibrosis

Cystic fibrosis (CF) is a rare disease (the prevalence in Spain is estimated to be 0.55 per 10,000 inhabitants [102]). Although not an infectious disease, per se, it is relevant here due to the chronicity of infections in these patients. The lungs of CF patients are generally infected by strains of *Haemophilus influenzae*, *P. aeruginosa, Stenotrophomonas maltophilia, S. aureus, Achromobacter* spp., *Burkholderia cepacia*, and *Mycobacterium abscessus*, usually resistant to many antimicrobials. Among them, the main pathogens in Spain are *S. aureus* and *P. aeruginosa* (chronically present in around half of the total patients), with the latter being significantly associated with worse pulmonary function. According to a Spanish multicenter study, 55% of *P. aeruginosa* isolates from CF patients were MRD, while 16% were extensively drug-resistant [103]. In addition, these microorganisms can form biofilms, which make antibiotic treatment difficult even when they are susceptible. Prolonged treatment with antimicrobials in adulthood also facilitates the settlement of nontuberculous mycobacteria, yeasts, and filamentous fungi, which further contribute to clinical disease. Colonization is associated with episodes of exacerbation of the respiratory symptoms and progressive deterioration of respiratory function, which is the leading cause of death in CF patients [104,105]. CF is the third most frequent indication for lung transplantation; however, the persistent infections common among CF patients relate to significant post-transplant morbidity and mortality, and in certain circumstances, these infections may even contraindicate transplantation. For this reason, the clearance of MDR infections in CF patients, particularly those caused by *P. aeruginosa*, is expected to have a remarkable impact on the life quality and expectancy of these patients.

### 6.2. Osteoarticular Infections

These are a particular form of deep, localized infections with poor response to antibiotic therapy. The diffusion of antibiotics into bone tissue is often low and is negatively affected by the presence of bacterial biofilms at the contact between bone and prosthetic material. Repeated surgeries to eradicate the biofilm mechanically, i.e., removing the implant and/or resecting the infected bone, and sometimes amputation, is the only infection control option [106]. Most of these infections are related to the hospital environment, such as those produced by multidrug-resistant *P. aeruginosa* and *S. aureus* resistant to methicillin, fluoroquinolones, and rifampicin. In these cases, treatment options are often limited to antibiotics of last resort, such as polymyxins which have been shown to have higher relapse rates and higher frequency of complications.

There are some recent examples of clinical cases (CF and osteoarticular infections) treated with phages that support the feasibility of this therapy (Table 2).

## 7. General Aspects of the Use of Endolysins as Antimicrobials

In addition to phages, their lytic enzymes have also been explored as antimicrobials [117]. Phage lysins are enzymes that mediate enzymatic cleavage of peptidoglycan either in the early stages of phage infection to assist in the injection of phage DNA or the late stages of infection, allowing the release of viral progeny by generalized lysis of the bacterium. Lysins are a promising new class of antibacterial agents that may also offer an answer to the problem of bacterial multidrug resistance. When added exogenously, they induce rapid osmotic lysis of Gram-positive bacteria by degradation of peptidoglycan with subsequent cell death. Their efficacy has also been extended to Gram-negative pathogens, which have a protective outer membrane, by protein engineering [118]. These endolysins are also called “enzybiotics” and have been shown to be safe, effective, fast-acting, and highly specific [119,120]. In addition, they weaken biofilms and have a low likelihood of causing resistance [119,120], and can be used either alone or in combination with traditional antibiotics [121]. Their potential applications are the same as those of phages, but they offer some advantages. For example, enzymes cannot propagate as phages do; therefore, their effect is directly dose-dependent and can be better controlled. In addition, they are not able to mobilize DNA, so horizontal gene transfer is avoided, and they should be easier to produce on an industrial scale given the prior experience with the production of heterologous therapeutic proteins.

Since peptidoglycan is present exclusively in bacteria and not in mammalian cells, the risk of cytotoxic effects to humans and animals is minimized. However, since they are proteins, they may induce an immune response. Indeed, in vitro and in vivo studies have shown that neutralizing antibodies can be generated upon repeated exposure to lysins. However, although these antibodies reduce the antibacterial activity of the enzybiotics, they do not completely neutralize them [122,123], which suggests that enzybiotics could be used repeatedly to treat the same bacterial infection [124]. The situation for the treatment of Gram-negative bacteria with lysins may be slightly different. Testing several non-engineered anti-Gram-negative lysins in vivo resulted in antibody production that did completely inhibit the in vitro bactericidal activity of the enzymes [125]. This suggests that further optimization must be taken into account when translating anti-Gram-negative enzybiotics to the clinic. Engineered endolysins against Gram-negative bacteria, nonetheless, have shown promising results in in vivo models of infection [126]. An additional major problem of many anti-Gram-negative enzybiotics is a lack of activity in the presence of human serum, probably due to their high cationicity, thus perhaps limiting their use for systemic treatment [127]. However, serum-active lysins against Gram-negative bacteria are already being developed by some companies and laboratories [126,128,129]. These advancements point out that, with appropriate experimental development, clinically valuable lysins will also be available in the medium term against Gram-negative pathogens. 

Systemic administration of enzybiotics may also release cellular debris from lysed bacteria, which may induce a proinflammatory response. These bacterial cell debris include lipopolysaccharides, (lipo)teichoic acids, and peptidoglycan through membrane fragmentation and can potentially lead to severe complications such as septic shock. However, allergic or severe inflammatory reactions have not been described so far upon administration of enzybiotics, even in clinical trials with human subjects [130,131]. The conclusions from some studies indicate that there should be an optimal dose of enzybiotics that is sufficient to kill the bacterial pathogen without additional fragmentation of the peptidoglycan layer and spreading of proinflammatory factors [122,123].

In recent years, the way has been paved for recombinant lysins to enter different phases in preclinical and clinical trials. Accelerated clinical advances and their high technical feasibility make them a good alternative therapy to replace or to be used in combination with conventional antibiotics in the short term, even with some authors pointing out that lysins may be able to enter the clinic in a shorter term than phages [132]. Some companies are already taking the initiative to conduct phase II and III clinical trials with enzybiotic-based products. An enzybiotic for cosmetic and topical application is already being marketed under the GladSkin brand, marketed by Micreos. Recently, human clinical trials (Phase I) have been conducted with two endolysins, iNtRON-N-Rephasin^®^ SAL200 (Tonabacase) [130] and Exebacase (CF-301) from ContraFect [133], against *S. aureus* bacteremia without observing any adverse effects. Tonabacase is currently in phase II [134], and Exebacase (CF-301) has completed phase II successfully and is in phase III to test its activity against bacteremia and endocarditis caused by *S. aureus* [135].

## 8. Global Phage Therapy Market

As discussed in this review, the application of phage therapy is experiencing a considerable boom in recent years. Since 2017, the global phage therapy market reached USD 567.9 million and growth expectations at compound annual growth rate (CAGR) are of 3.9% for the period from 2018 to 2026 [136]. In 2016, USA had a 37% share of the global market and became the largest market in the use of phages [137]. Europe is the second-largest market due to the wide application of phages in the food and environmental fields. The particular cases of Georgia, Poland, and Belgium are especially relevant because they are pioneer countries in having taken the step from basic research to the market. It seems most probable that this market will experience growth in the short term, supported by heavy investment in biotechnology infrastructures, the evolution of agricultural practices and their corresponding regulation, and foreseeable government initiatives favoring the use of these technologies (or at least discouraging the massive use of antibiotics) in different sectors, including that of human health, due to the antibiotic resistance crisis [138]. In this regard, the most promising market for phage therapy would be the treatment of infections caused by MDR bacteria. Estimates are that this will globally represent more than USD 13.8 billion a year by 2027 [139].

However, some potential hurdles may still be found in the way towards a successful market entrance, such as the already mentioned regulatory issues (Section 4), the quality of the results from future phage therapy clinical trials (Section 5), or questions about phage patentability (although patents granted for phage products do already exist) [132,140]. The more “canonical” nature of enzybiotics, similar to other therapeutic proteins, is an additional reason to think that they may make their market entrance in a shorter term than full phages, following the traditional avenue through intellectual property protection of the molecules and conduction of clinical trials (Section 7). However, provided that reasonable regulatory measures come into place, therapy with full virions, especially in its personalized version, may be available to patients first, although its path to proper marketability may be longer.

In this context and given that large pharmaceutical companies have practically stopped investing in new antimicrobial compounds, many scientific startups are focusing on studying and characterizing phages and their products for their therapeutic application. These new companies are located mainly in the USA, India, Korea, Canada, and some European countries. In Spain, the first company focused on the development of endolysins (Telum Therapeutics) has recently been created [141]. An optimistic short-term future can also be envisioned for these developments if we consider the latest investments taking place in the area of alternatives to antibiotics, such as the BioNtech acquisition of PhagoMed [142], or the high investment rounds to support new endolysins search platforms, as in Micreos [143].

## 9. Final Conclusions

Bacterial resistance is a major global threat and a leading cause of mortality worldwide. The lack of effective antibiotics against MDR strains urgently requires therapeutic alternatives, and phages or their derivatives can be a promising approach with sufficient scientific and technological know-how in place. However, the lack of both specific regulation for their clinical use and public awareness stands out, in our opinion, as the major hurdles to be presently faced. With this review, we would like to emphasize the advantages of phages as therapeutic tools and how they can be used to combat MDR bacteria. Moreover, despite some foreseeable drawbacks also covered here, phages are nowadays the only short-term solution for many patients. Taking this into account, a suitable regulatory initiative from the competent authorities would be welcomed in Spain in the short term. Both scientific literature and the lead examples of other countries prove that phage therapy is already mature to be translated into a regulation that must ease, protect, and help developing the phage therapy market and infrastructure.

## Figures and Tables

**Figure 1 microorganisms-10-00717-f001:**
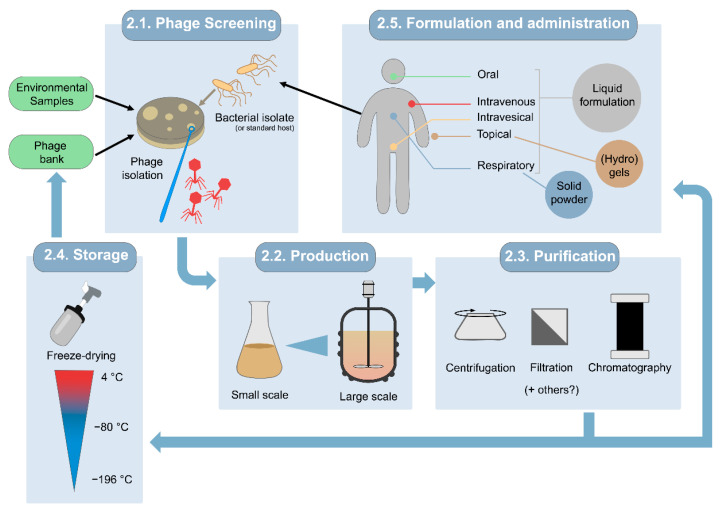
Steps in preparation of phage suspensions suitable for phage therapy, including screening, propagations, purification, storage, and formulation.

**Table 1 microorganisms-10-00717-t001:** Ongoing clinical trial examples involving phage therapy.

Disease	Pathogen(s)	Treatment	Status	References
Diabetic foot ulcers	*Staphylococcus aureus*	Topical phage cocktail	Not yet recruiting (expected start date: June 2022)	NCT02664740
Invasive infection in patients with inactive Crohn’s disease	*E. coli*	Oral phage cocktail	Recruiting (estimated completion: June 2023)	NCT03808103
Chronic airway infection in cystic fibrosis patients	*P. aeruginosa*	Nebulized phage therapy	Recruiting (estimated completion: December 2022)	NCT04684641
Diabetic foot ulcers	*P. aeruginosa, S. aureus* and/or *Acinetobacter baumannii*	Topical phage cocktail	Recruiting (estimated completion: December 2021	NCT04803708
Prosthetic joint infections	Several pathogens	Combined antibiotic/personalized phage therapy	Not yet recruiting (estimated start date: October 2022)	NCT04787250
Chronic airway infection in cystic fibrosis patients	*P. aeruginosa*	Nebulized phage cocktail	Not yet recruiting	NCT05010577
Wound infections in burned patients	*S. aureus, P. aeruginosa* or *Klebsiella pneumoniae*	Topical phage cocktail	Not yet recruiting (estimated start date: January 2022)	NCT04323475
Pressure injury infections	*S. aureus, P. aeruginosa, K. pneumoniae*	Topical phage cocktail in combination with antibiotics	Not yet recruiting (estimated start date: January 2022)	NCT04815798
Urinary tract infections	*E. coli* or *K. pneumoniae*	Personalized phage therapy administered through intravenous or intravesical route	Recruiting (estimated completion: September 2023)	NCT04287478
Tonsillitis	Several pathogens	Nebulized phage cocktail	Phase 3. Active, not recruiting (estimated completion: December 2024)	NCT04682964
Chronic airway infection in cystic fibrosis patients	*P. aeruginosa*	Inhaled phage cocktail	Recruiting (estimated completion: March 2022)	NCT04596319

**Table 2 microorganisms-10-00717-t002:** Examples of clinical cases treated with phage therapy.

Disease	Pathogen(s)	Treatment	Outcome	References
CF with chronic MDR lung infection	*Achromobacter* *xylosoxidans*	Inhalation, orally	Dyspnea resolved and cough reduced. Lung function improved	[107]
CF with disseminated infection, lung transplantation	*M. abscessus*	Intravenous	Sternal wound closure, improved liver function, substantial resolution of infected skin nodules	[108]
CF with MDR pneumonia, persistent respiratory failure, and colistin-induced renal failure	*P. aeruginosa*	Intravenous	Pneumonia clinically resolved, no sputum production, return to baseline renal function, white blood cell count normalized	[109]
CF with persistent lung infection, lung transplantation	*A. xylosoxidans*	Inhalation	Respiratory condition improved; sputum cultures positive but with low bacteria concentration	[110]
Lung transplant recipient patients with MRD resistant infections	*P. aeruginosa* and *Burkholderia dolosa*	Intravenous, inhalation	Two patients were discharged from the hospital off ventilator support. A third patient infection relapsed and died	[111]
COPD with drug-resistant pneumonia	*A. baumannii*	Inhalation	Sputum/ blood and bronchoalveolar lavage fluid negative, restoration sinus rhythm, lung function improved	[112]
Prosthesis infection	*S. aureus*	Local	Bacteria removed, rapid healing	[113]
Osteomyelitis	*P. aeruginosa*	Local	No clinical signs of persistent infection	[114]
Infection of the right knee and chronic osteomyelitis of the femur after injury	*P. aeruginosa*	Local	No pain, soft tissue at the surgical site unremarkable, mobility satisfactory	[115]
Osteomyelitis of the distal phalanx	*S. aureus*	Local	The ulcer healed, re-ossification of the distal phalanx, erythema and edema decreased	[116]
Fracture-related infection	*K. pneumoniae*	Local	Skin graft vascularized and viable, the sinus tract closed and dry, pus no longer discharged from the pin sites of the external fixator, restored muscle function	[20]

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
