# Peer review of "Essential Topics for the Regulatory Consideration of Phages as Clinically Valuable Therapeutic Agents: A Perspective from Spain"

_microorganisms, 2022, doi:10.3390/microorganisms10040717_

Round 1
Reviewer 1 Report
In the article “Essential Topics for the Regulatory Consideration of Phages and Endolysins as Clinically Valuable Therapeutic Agents: a perspective from Spain” a revision is made for several aspects of using phage therapy. However, even though the Endolysins term is part of the manuscript title, there is only one section dedicated to this. In this reviewer opinion, the title should be reformulated for phages only, nevertheless in the manuscript this section can be maintained since these proteins are in fact being explored and are phage proteins.
This is a value revision of the hurdles of using/applying phage therapy in Spain, additionally the sections that explored phage therapy are important for any country that intends to apply this therapy. The phage global therapy market is a good section, not usually seen in other revisions.
Some major/minor concerns with the manuscript that arise from reading it are:
Q1. Line 62 – which is repeated. Please reformulate the sentence.
Q2. Line 75 – It is possible to explain why the PAS approach is the safest? It is also the most efficient? It is safest than using only phages or only antibiotics? And why?
Q3. It would be a good addition to the manuscript a schematic version of section 2 (line 195). As the manuscript is extensive it helps the reader to go through this pipeline easier.
Q4. Throughout the manuscript there are references to bacteriophages or phages. To maintain consistency only one form should be selected and used. Please check the manuscript for this.
Q5. Line 378 – As GMP appears for the first time in the manuscript it should be in full form first and then the acronym can be used for the rest of the text.
Q6. Table 2 – It is possible to add the outcome of the treatments to this table? It would be a good addition to show the potential of using phage therapy.
Author Response
In the article “Essential Topics for the Regulatory Consideration of Phages and Endolysins as Clinically Valuable Therapeutic Agents: a perspective from Spain” a revision is made for several aspects of using phage therapy. However, even though the Endolysins term is part of the manuscript title, there is only one section dedicated to this. In this reviewer opinion, the title should be reformulated for phages only, nevertheless in the manuscript this section can be maintained since these proteins are in fact being explored and are phage proteins.
Thank you for the comment, we agree with this suggestion and we removed “endolysins” from the title.
This is a value revision of the hurdles of using/applying phage therapy in Spain, additionally the sections that explored phage therapy are important for any country that intends to apply this therapy. The phage global therapy market is a good section, not usually seen in other revisions.
Some major/minor concerns with the manuscript that arise from reading it are:
Q1. Line 62 – which is repeated. Please reformulate the sentence.
Done
Q2. Line 75 – It is possible to explain why the PAS approach is the safest? It is also the most efficient? It is safest than using only phages or only antibiotics? And why?
The combined use of phages and antibiotics (PAS) results in a synergistic effect that allows a reduction in the concentration of both antimicrobials. Additionally, a fraction of bacteria would turn out resistant to the phage and consequently, it loses their ability to resist to antibiotic and even in some cases, also lose virulence. In summary, antibiotic resistance and phage resistance are minimized and the efficacy and safety of the treatment is maximum.
The text was modified accordingly.
Q3. It would be a good addition to the manuscript a schematic version of section 2 (line 195). As the manuscript is extensive it helps the reader to go through this pipeline easier.
Done
Q4. Throughout the manuscript there are references to bacteriophages or phages. To maintain consistency only one form should be selected and used. Please check the manuscript for this.
Done
Q5. Line 378 – As GMP appears for the first time in the manuscript it should be in full form first and then the acronym can be used for the rest of the text.
Done
Q6. Table 2 – It is possible to add the outcome of the treatments to this table? It would be a good addition to show the potential of using phage therapy.
Done
Reviewer 2 Report
This review article is dedicated to a very important topic. Phage therapy has an undoubted future, but the use of phages currently faces many regulatory challenges.
The article covers wide range of aspects, each statement is referenced and scientifically based on both the possible application and possible side effects and problems. In addition, there is an overview of the use of endolysins as antimicrobials. I fully agree with the author assessment in the "Final conclusions" section.
Typographical error on line 592.
Author Response
This review article is dedicated to a very important topic. Phage therapy has an undoubted future, but the use of phages currently faces many regulatory challenges.
The article covers wide range of aspects, each statement is referenced and scientifically based on both the possible application and possible side effects and problems. In addition, there is an overview of the use of endolysins as antimicrobials. I fully agree with the author assessment in the "Final conclusions" section.
Typographical error on line 592.
Done